# Operationally classical simulation of quantum states

Gabriele Cobucci [1,6], Alexander Bernal [1,2,6], Martin J. Renner [3,4,5] & Armin Tavakoli [1]

A classical state-preparation device cannot generate states in relative superposition. We introduce classical models in which devices that are individually unable to generate states with relative superposition can be stochastically coordinated to simulate sets of quantum states. These models have natural operational interpretation in prepare-and-measure scenarios and they can account for many non-commuting quantum state sets. We develop systematic methods both for classically simulating quantum sets and for showing that no such simulation exists, thereby certifying quantum coherence. In particular, we determine the exact noise rates required to classically simulate the entire state space of quantum theory. We also reveal connections between the operational classicality of sets and the well-known fundamental concepts of joint measurability and Einstein-Podolsky-Rosen steering. Here, we present an avenue to understand how and to what extent quantum states defy generic models based on classical devices, which also has relevant implications for quantum information applications.

The ability of quantum theory to break the constraints of classical physics is essential both for conceptually understanding the theory and for its many emerging applications in information technology. Perhaps the most fundamental non-classical feature of quantum theory are superposition states. The superposition principle directly leads to the notion of coherence in sets of quantum states and its formal characterisation has received much interest in recent times[1]. However, whether and to what extent a quantum set is coherent hinges on the power we grant to the models that we call classical. In other words, when trying to simulate quantum states, what should classical models be allowed to do?

The most studied approach to quantum coherence is to assume that there is a special basis to which classical models are restricted[2]. Only the states that are diagonal in this basis are considered classical. This means e.g. that the qubit states $|\pm\rangle = \frac{|0\rangle \pm |1\rangle}{\sqrt{2}}$ are coherent if the

classicality basis is chosen as $\{|0\rangle, |1\rangle\}$, but they are not coherent if we switch reference frame to the basis $\{|+\rangle, |-\rangle\}$. Thus, this approach views coherence as a relative property. In many systems there are good arguments for introducing such a privileged basis. Nevertheless, this constitutes in general a limitation on models that would reasonably be considered classical.

A less restrictive approach is to consider as classical any state preparation device which emits states without relational coherences. In other words, the states created by this device must all be diagonal in the same basis, but the basis can be arbitrary. This ensures that classicality no longer is associated with any privileged basis, but consequently, any single quantum state $|\psi\rangle$ is classical because it is always diagonal in some basis. Therefore, such absolute quantum coherence can only be found in sets of quantum states, i.e. a set of states $\{\rho_1, \rho_2, ...\}$, that cannot be collectively diagonalised. It is an introductory textbook fact that collective diagonalisation is possible

[1]Physics Department and NanoLund, Lund University, Lund, Sweden. [2]Instituto de Física Teórica, IFT-UAM/CSIC, Universidad Autónoma de Madrid, Madrid, Spain. [3]ICFO - Institut de Ciencies Fotoniques, The Barcelona Institute of Science and Technology, Castelldefels, Barcelona, Spain. [4]University of Vienna, Faculty of Physics and VDSP, Vienna Center for Quantum Science and Technology (VCQ), Vienna, Austria. [5]Institute for Quantum Optics and Quantum Information (IQOQI), Austrian Academy of Sciences, Vienna, Austria. [6]These authors contributed equally: Gabriele Cobucci, Alexander Bernal. ✉e-mail: armin.tavakoli@fysik.lu.se

if and only if all the states commute. The extent to which a quantum set fails to commute has also been quantified[3].

However, commutation poses a very strong constraint on quantum sets. Consider for instance a set of two states corresponding to the positive eigenstates of Pauli operators $\sigma_X$ and $\sigma_Z$, but subject to noise, so their visibility is $v \in [0, 1]$. We can write the two states as $\rho_1 = v|0\rangle\langle 0| + \frac{1-v}{2}\mathbb{1}$ and $\rho_2 = v|+\rangle\langle +| + \frac{1-v}{2}\mathbb{1}$. These commute only when $v = 0$. The same would hold if we would rotate $\rho_2$ arbitrarily close to $\rho_1$. Hence, regardless of how close and noisy the states become, they commute only when they are identical. Thus, the set $\{\rho_1, \rho_2\}$ is coherent almost always. Nevertheless, for small enough $v > 0$, the set would not be expected to be a resource in any non-contrived quantum technology application. This may suggest that commutation is insufficient for capturing the full scope of models that can reasonably be called classical.

Here, we introduce an avenue for classical models that simulate sets of quantum states. The idea that underpins this approach is that the no-superposition requirement is imposed on each of the actual state-preparation devices used in the simulation. This contrasts the conventional way of asking whether the considered quantum states commute. Specifically, we suppose that we have access to many different state-preparation devices and that each of these only emits states diagonal in some basis. We then let a random variable govern the choice of which device to call upon when trying to simulate the quantum set. This procedure is schematically illustrated in Fig. 1. We will show that such operationally classical models are strictly more powerful because they can simulate many quantum sets that do not commute.

We now describe the organisation of our article, with a brief summary given in the Table 1.

In Section "Results" "Classical models", we introduce operationally classical models and identify their basic properties. We discuss how the failure of classical models for sets of quantum states plays a role for the quantum prepare-and-measure scenario analogous to the role played by entanglement in the Bell nonlocality scenario. Next, we set out to answer the central question, namely that of characterising the quantum sets that admit classical models. In Section "Results" "Simulation of quantum sets", we develop both analytical and numerical methods to find classical models for quantum sets. Of particular conceptual interest is to consider classical simulation of all pure quantum states for a given Hilbert space dimension $d$. We determine the precise amount of noise that must be added to $d$-dimensional quantum theory in order to render it classical. In Section "Results" "Detecting absolute quantum coherence", we take the opposite perspective and systematically derive criteria for showing that no classical model exists. Our criteria are computable with standard means and they are designed to be testable in realistic experimental settings. In Section "Results" "Foundational aspects", we show that our notion of classicality for states has connections to well-known fundamental concepts in quantum theory. We show that it implies a particular form of joint measurability[4] and that criteria for testing Einstein-Podolsky-Rosen steering[5] can be transformed into criteria for testing the classicality of quantum sets. In Section "Discussion" we provide a concluding discussion on the broader relevance of the concepts and results.

## Results
### Classical models
Consider a set $\mathcal{E} = \{\rho_x\}_{x=1}^m$ comprised of $m$ quantum states on a $d$-dimensional Hilbert space. The set is commonly considered classical if no pair of states are coherent with respect to each other. Equivalently, there exists a basis in which all the states are diagonal and this occurs if and only if every pair of states commute, i.e. if $[\rho_x, \rho_{x'}] = 0 \,\forall x, x'$. We build on this reasoning to propose models that are classical in the sense that joint diagonalisation holds for every preparation device used by the model. The central observation is that since commutation is a basis-independent notion, it permits a classical state-preparation device to generate states that are collectively diagonal in any desired basis. We therefore consider a model that has access to many different devices of this type; see Fig. 1. Let us label these different devices by $\mathcal{P}_1, \mathcal{P}_2, \ldots$ etc. We may allow even an uncountably infinite number of them, $\{\mathcal{P}_\lambda\}_\lambda$. Here, each $\lambda$ is associated with the orthonormal basis of $d$-dimensional Hilbert space in which all states emitted from $\mathcal{P}_\lambda$ are diagonal. We label the set of states emitted from $\mathcal{P}_\lambda$ by $\{\xi_{z,\lambda}\}_z$, where $z$ indicates the specific output state. Since the device is classical, it cannot generate relational coherence, and hence all emitted states commute, $[\xi_{z,\lambda}, \xi_{z',\lambda}] = 0 \,\forall z, z'$. Moreover, we may exploit pre- and post-processing of all these devices. Specifically, the pre-processing amounts to stochastically choosing which device to use in each round of the experiment. This is represented by a probability density function $q(\lambda)$ which satisfies the standard conditions of non-negativity, $q(\lambda) \geq 0$, and normalisation, $\int d\lambda\, q(\lambda) = 1$. The post-processing amounts to stochastically wiring the output states from the different devices. That is, when aiming to generate the quantum state $\rho_x$, we can with some conditional probability density $p(z|x, \lambda)$ select the output state

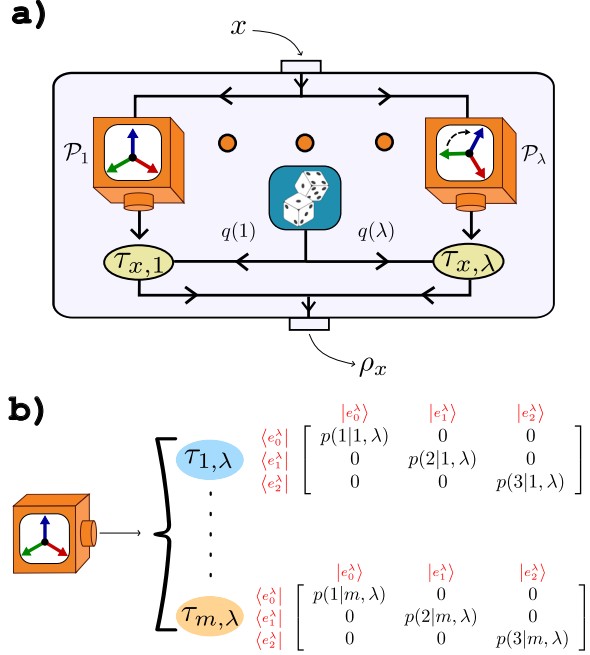

**a)**

**b)**

**Fig. 1 | Classical models for quantum sets.** For a given set of quantum states $\{\rho_x\}_x$, we ask if it be modelled only using classical devices. **a** Many independent classical state-preparation devices $\mathcal{P}_1, \mathcal{P}_2, \ldots$ are called stochastically via a probability density function $q(\lambda)$. **b** The classicality of $\mathcal{P}_\lambda$ means that its emitted states $\{\tau_{x,\lambda}\}_x$ commute, i.e. there exists a basis $\{|e_k^\lambda\rangle\}_k$ in which all $\{\tau_{x,\lambda}\}_x$ are diagonal. This is illustrated for a three-dimensional example.

**Table 1 | Organisation of the paper**

| Section | Main content |
|---|---|
| II A | Introducing operationally classical models |
| II E | How to design classical models for quantum sets |
| II F | Criteria for ruling out classical simulability |
| II G | Relations to fundamental concepts and their applications |
| III | Discussion and interpretation for quantum information |

$\xi_{z,\lambda}$ from $\mathcal{P}_\lambda$. Thus, if it is indeed possible to simulate the set of quantum states, $\mathcal{E}$, by the above classical means, each state in the set admits the form

$$\rho_x = \int d\lambda\, q(\lambda) \int dz\, p(z|x,\lambda)\, \xi_{z,\lambda},\ \forall x. \tag{1}$$

It is useful to note that the post-processing can be eliminated without loss of generality. We need only to define $\tau_{x,\lambda} \equiv \int dz\, p(z|x,\lambda)\xi_{z,\lambda}$ and observe that $\tau_{x,\lambda}$ is a valid state diagonal in the same basis as $\{\xi_{z,\lambda}\}_z$. Therefore, we can consider that $\mathcal{P}_\lambda$ directly emits the commuting set $\{\tau_{x,\lambda}\}_{x=1}^m$. We have arrived at the definition of classical models for sets of quantum states.

**Definition 1.** (Classical models) Let $\mathcal{E} = \{\rho_x\}_{x=1}^m$ be a set of $d$-dimensional quantum states. The set is called classically simulable if it can be written as

$$\rho_x = \int d\lambda\, q(\lambda)\, \tau_{x,\lambda},\ \forall x, \tag{2}$$

for some probability density function $q(\lambda)$ and some set of states $\{\tau_{x,\lambda}\}$ where $[\tau_{x,\lambda}, \tau_{x',\lambda}] = 0\ \forall x, x', \lambda$.

We now identify some elementary features of classical models. Firstly, notice that commuting sets $\{\rho_x\}_x$ correspond to the special case in which $q(\lambda)$ is deterministic, i.e. when only a single device $\mathcal{P}_\lambda$ is employed in the simulation. It therefore follows trivially that any single-state set ($m = 1$) is classically simulable. Secondly, any non-deterministic choice of $q(\lambda)$ must introduce mixedness in the simulation. Therefore, if $\mathcal{E}$ consists only of pure states, classical simulability reduces to commutation. Thirdly, unlike the set of commuting sets of states, the set of classically simulable sets of states is convex by construction. Indeed, given two classically simulable sets $\mathcal{E} = \{\rho_x\}_x$ and $\mathcal{E}' = \{\sigma_y\}_y$, then $\{p\rho_x + (1-p)\sigma_y\}_{x,y}$ is classically simulable for $p \in [0, 1]$ (see SM for a proof).

We label that set by $\mathcal{S}$, with the specific number of states ($m$) and the dimension ($d$) left implicit. Fourthly, if $\mathcal{E} \in \mathcal{S}$ then also any sub-set $\mathcal{E}' \subset \mathcal{E}$ is classically simulable simply by discarding states from the simulation of $\mathcal{E}$.

### Interpretation in the prepare-and-measure scenario

The main idea leading up to Definition 1 is that while individual classical preparation devices cannot generate superpositions, one can stochastically coordinate many of these devices with the aim of enhancing their simulation capabilities beyond merely quantum states that commute. In making this definition, we required that the stochastic variable ($\lambda$) is independent of the label of the state we want to simulate ($x$). If $\lambda$ is allowed to depend on $x$ then Definition 1 becomes trivial. This independence can be physically motivated by a standard freedom-of-choice assumption when selecting $x$ and viewing $\lambda$ as a pre-programming of the device. This pre-programming can be created elsewhere and then distributed to the device that implements the classical simulation. This is analogous to the justification of local hidden variable models in Bell nonlocality tests.

Once viewing $\lambda$ as distributed to the device, a natural connection can be made between our notion of classical models and the prepare-and-measure scenario. The prepare-and-measure scenario features a sender, Alice, and a receiver, Bob. They are allowed to share a classical random variable, $\lambda$, which is distributed to them. Alice selects an input, $x$, and encodes it into a message that is sent to Bob. Bob selects a classical input $y$ and maps the message into his output, $b$. If the parties operate without quantum resources and the message is limited to a $d$-valued alphabet, their correlation statistics becomes[6,7]

$$p(b|x,y) = \int d\lambda\, q(\lambda) \sum_{m=1}^d p_A(m|x,\lambda) p_B(b|m,y,\lambda), \tag{3}$$

where $q(\lambda)$ is the density function of $\lambda$, and $p_A$ and $p_B$ are the stochastic response functions of Alice and Bob. In a quantum model, the message is replaced with a $d$-dimensional state and Bob applies quantum measurements to obtain $b$. It is well-known that such semi-device-independent quantum correlations in general are incompatible with the model in Eq (3)[8,9].

Consider now a quantum model where Alice prepares non-commuting quantum states $\{\rho_x\}_x$, but assume that this set of states nevertheless admits a classical model in the sense of Definition 1. Then, it follows that any quantum correlation statistics, $p(b|x,y) = \mathrm{Tr}\left(\rho_x M_{b|y}\right)$, regardless of Bob's choice of measurements $\{M_{b|y}\}$, is compatible with the model in Eq (3). The reason for this is that once a decomposition of the form (2) exists, Bob can use his knowledge of the shared variable $\lambda$ to know the basis in which $\tau_{x,\lambda}$ is diagonal, $\{|e_k^\lambda\rangle\}_k$, independently of Alice's choice of $x$ (see Fig. 2). Hence, from Bob's point of view, he is always receiving a diagonal state, which is effectively equivalent to a stochastic classical message. This is illustrated in Fig 2. Formally, the correlations obtained from a quantum model in which the states admit a classical model becomes

$$p(b|x,y) = \int d\lambda\, q(\lambda) \sum_{m=1}^d p(m|x,\lambda)\left\langle e_m^\lambda | M_{b|y} | e_m^\lambda \right\rangle, \tag{4}$$

which takes the form of Eq (3) if we define $p_B(b|m,y,\lambda) = \langle e_m^\lambda | M_{b|y} | e_m^\lambda \rangle$.

Hence, the failure of our classical models for $\{\rho_x\}_x$ are in the prepare-and-measure scenario analogous to the failure of separable models (entanglement) for the bipartite state used in the Bell nonlocality scenario. Both our models and separable models concern the nature of the quantum entities that underly the observed statistics in the respective scenarios. In view of that, it is operationally natural that the failure of such a model is a necessary but in general not a sufficient condition for quantum correlations in the respective black-box scenarios.

### Example: simulation of noisy BB84 states

A central consequence of Definition 1 is that some sets of quantum states that do not commute nevertheless are classically simulable. We showcase this through a simple example based on the set of qubit states used in the BB84 quantum key distribution protocol. Consider the set of four qubits, $\mathcal{E} = \{\rho_0, \rho_1, \rho_+, \rho_-\}$, corresponding to the noisy eigenstates of the Pauli $\sigma_X$ and $\sigma_Z$ operators. These are $\rho_x = v|x\rangle\langle x| + \frac{1-v}{2}\mathbb{1}$ for $x \in \{0, 1, +, -\}$ for some visibility $v \in [0, 1]$. Only do all pairs of states commute when $v = 0$, but we now show that they are classically simulable in the range $0 \le v \le \frac{1}{\sqrt{2}}$. To show this, we need only to use two classical devices, $\mathcal{P}_1$ and $\mathcal{P}_2$, and we call them with equal probability, $q(1) = q(2) = \frac{1}{2}$. Let $\mathcal{P}_1$ generates two orthogonal (i.e. commuting) states $\{|\varphi\rangle, |\varphi_\perp\rangle\}$. Select them as $|\varphi\rangle = \cos(\frac{\pi}{8})|0\rangle + \sin(\frac{\pi}{8})|1\rangle$ and $|\varphi_\perp\rangle = \sin(\frac{\pi}{8})|0\rangle - \cos(\frac{\pi}{8})|1\rangle$. Similarly, let $\mathcal{P}_2$ generate the two orthogonal states $\{|\chi\rangle, |\chi_\perp\rangle\}$ where $|\chi\rangle = \cos(\frac{\pi}{8})|0\rangle - \sin(\frac{\pi}{8})|1\rangle$ and $|\chi_\perp\rangle = \sin(\frac{\pi}{8})|0\rangle + \cos(\frac{\pi}{8})|1\rangle$. To see that the classical simulation succeeds, we need only to select $v = \frac{1}{\sqrt{2}}$ and note that all four noisy BB84 states are recovered by mixing states from $\mathcal{P}_1$ and $\mathcal{P}_2$,

$$\begin{aligned}\rho_0 &= \frac{1}{2}\varphi + \frac{1}{2}\chi, &\rho_+ &= \frac{1}{2}\varphi + \frac{1}{2}\chi_\perp, \\ \rho_1 &= \frac{1}{2}\varphi_\perp + \frac{1}{2}\chi_\perp, &\rho_- &= \frac{1}{2}\varphi_\perp + \frac{1}{2}\chi,\end{aligned} \tag{5}$$

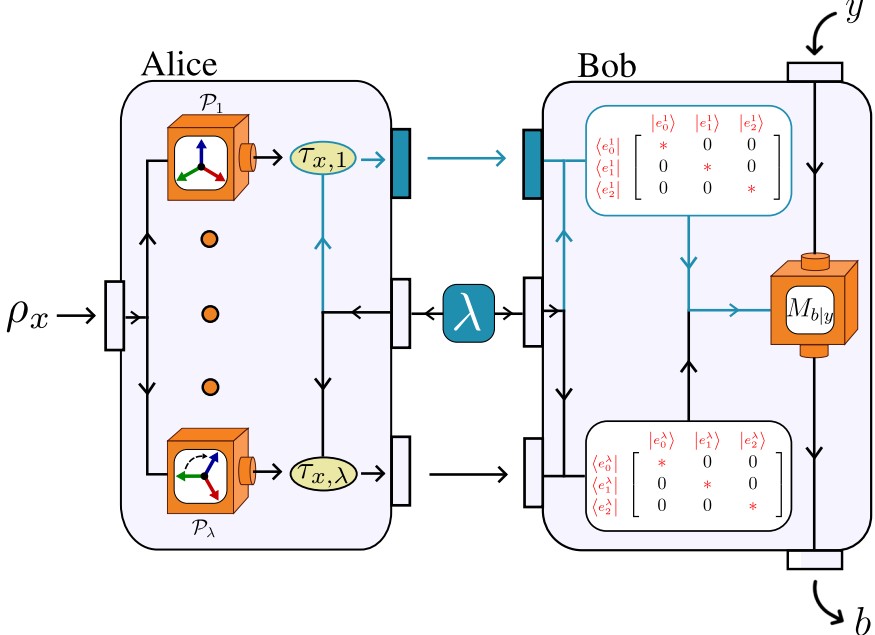

**Fig. 2 | Classical models in prepare-and-measure scenarios.** $\lambda$ is distributed between Alice and Bob. The existence of a classical model for Alice's preparations allows Bob to know the basis in which her quantum states are diagonal, independently of her input $x$. Then, any quantum correlation statistics $p(b|x, y)$ can be simulated sending a classical message limited to a $d$-valued alphabet.

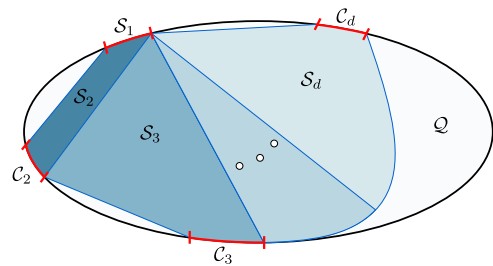

**Fig. 3 | Quantum vs classical sets.** The sets of quantum states that admit a classical model with complexity $r \in \{1, ..., d\}$ are convex and form a nested structure, leading up to the full collection of classical sets $\mathcal{S}$. Commuting sets $\mathcal{C}_r$ in dimension $r$ are represented as the red boundaries in the figure. The white region represents quantum sets that cannot be classically simulated, while the black boundary denotes the space of non-commuting pure states sets.

where for any state $|\psi\rangle$, we define $\psi = |\psi\rangle\langle\psi|$. By the convexity of $\mathcal{S}$, it follows that $\mathcal{E}$ is classically simulable also for the range $0 \le v \le \frac{1}{\sqrt{2}}$.

Interestingly, it is known that when $v > \frac{1}{\sqrt{2}}$ the four noisy BB84 states can be used in a random access code task to produce quantum correlations in the semi-device-independent prepare-and-measure scenario that defy the model in Eq (3)[6]. Since the falsification of a classical model for the noisy BB84 states is a necessary condition for observing such correlations, it follows from our above example that there exists no concievable task in the standard (black-box) prepare-and-measure scenario that can better harvest quantum correlations from these states than does the random access code. In other words, our classical model for the noisy BB84 set can be viewed as the analogy of an optimal local hidden variable model but now for the prepare-and-measure scenario.

## Complexity of classical models

Once a set of quantum states is found to admit a classical model, it is natural to ask how complex the model is. We propose to use the dimensionality of the devices $\mathcal{P}_\lambda$ as qualitative measure of the complexity of the classical model. This dimensionality is a parameter $r \in \{1, 2, ..., d\}$ and it means that each device is restricted not only to emit commuting states but also to emit states supported only in some $r$-dimensional subspace of the $d$-dimensional Hilbert space. For $\mathcal{P}_\lambda$, we call that $r$-dimensional subspace $\Pi_\lambda$. Note that if $r = d$ the complexity is maximal; we therefore have $\Pi_\lambda = \mathbb{1}$ and hence all commuting states are allowed, as in Definition 1 for generic classical models. Thus, the definition of classical simulation complexity can be seen as a generalisation of Definition 1 which distinguishes the classical simulations that have lower-than-maximal complexity ($r < d$).

**Definition 2.** (Classical simulation complexity) Let $\mathcal{E} = \{\rho_x\}_{x=1}^m$ be a classically simulable set of $d$-dimensional quantum states. The simulation complexity is

$$\begin{aligned}
r_C(\mathcal{E}) \equiv \min_{q,\tau} \{r : \rho_x = \int d\lambda \, q(\lambda) \, \tau_{x,\lambda}, \\
\text{where } \exists \Pi_\lambda \text{ s.t. } \Pi_\lambda^2 = \Pi_\lambda, \\
\text{tr}(\Pi_\lambda) = r, \, \tau_{x,\lambda} = \Pi_\lambda \tau_{x,\lambda} \Pi_\lambda \, \forall x, \\
[\tau_{x,\lambda}, \tau_{x',\lambda}] = 0 \, \forall x, x'\}
\end{aligned}$$

(6)

where the minimisation is evaluated over the states $\{\tau_{x,\lambda}\}$ and the probability density function $q(\lambda)$, where $\lambda$ runs over all possible $r$-dimensional classical devices.

We denote by $\mathcal{S}_r$ the set of states with classical simulation complexity $r$ and we refer to any $\mathcal{E} \in \mathcal{S}_r$ as an $r$-*simulable* set. The sets of $r$-simulable states are convex and have the nested structure

$$\mathcal{S}_1 \subset \mathcal{S}_2 \subset \ldots \subset \mathcal{S}_d = \mathcal{S}. \tag{7}$$

The lowest simulation complexity is particularly simple and is comprised by all single-state sets: $\rho_1 = \rho_2 = \ldots = \rho_m$. To see this, notice that when $r = 1$ we write $\Pi_\lambda = |\varphi_\lambda\rangle\langle\varphi_\lambda|$ and hence the only state compatible with the constraint $\tau_{x,\lambda} = \Pi_\lambda \tau_{x,\lambda} \Pi_\lambda$ is $\tau_{x,\lambda} = |\varphi_\lambda\rangle\langle\varphi_\lambda| \equiv \tau_\lambda$. Thus, $\rho_x = \int d\lambda q(\lambda) \tau_\lambda \, \forall \, x$. Qualitative relations between the discussed sets are illustrated in Fig. 3.

## Simulation of quantum sets

Equipped with the classical models, the central challenge is to characterise the quantum sets that admit a simulation. In this section, we develop classical models for arbitrary given quantum sets. We will approach this first by analytical and then by numerical means.

Consider that we are given some set of $d$-dimensional pure states $\{|\psi_x\rangle\}_x$ where at least one pair of states is non-commuting. The whole set is then exposed to isotropic noise of visibility $v \in [0, 1]$, thus taking the form

$$\mathcal{E} = \{\rho_x\}_{x=1}^m, \quad \text{where } \rho_x = v|\psi_x\rangle\langle\psi_x| + \frac{1-v}{d}\mathbb{1}. \tag{8}$$

This is a frequently studied class of sets in quantum theory and particularly so in the context of quantum information where isotropic noise often represents a relevant approximation for experiments. Our goal is to classically simulate these sets. Note that for $v = 0$ all states are identical, so $\mathcal{E} \in \mathcal{S}_1 \subset \mathcal{S}$ is trivially classical. On the contrary, for $v = 1$ the states are pure and in general non-commuting so $\mathcal{E} \notin \mathcal{S}$ is not classical. Hence, the relevant question is to determine bounds on the critical value of the visibility, below which the set is simulable and above which it is not.

We begin with presenting a general sufficient condition for classical simulability that applies to any set of the form (8), i.e. it is valid independently of the choice of $\{|\psi_x\rangle\}_x$.

**Result 1.** (Simulation model) Consider any set of pure states mixed with isotropic noise, as given in Equation (8). The set is classically simulable for visibilities

$$v \leq \frac{H_d - 1}{d - 1}, \tag{9}$$

where $H_n = \sum_{k=1}^n \frac{1}{k}$ is the Harmonic number. Moreover, it is $r$-simulable when $v \leq (H_r - 1)/(d - 1)$.

**Proof.** We present the main ideas used in the proof and refer to SM for details. Following Definition 2 we construct an explicit model that uses uncountably many devices $\mathcal{P}_\lambda$. To this end, let us consider any unitary transformation of the computational basis, $\{U|i\rangle\langle i|U^\dagger\}_{i=1}^d$. We select the subspace corresponding to the $r$ first elements of this rotated basis, $\Pi_\lambda \equiv \Pi_U = \sum_{i=1}^r U|i\rangle\langle i|U^\dagger$. We simulate each $\rho_x$ in the set by averaging over the Haar measure the state $\tau_{x,\lambda}$, which is chosen to be the basis element in $\{U|i\rangle\langle i|U^\dagger\}_{i=1}^r$ that has the largest overlap with the pure state $|\psi_x\rangle$ in Equation (8). Since for each $\lambda$, equivalently $U$, all $\tau_{x,\lambda}$ are elements of the same common basis, they trivially commute. The Haar integral yielding the simulation is computed using the techniques presented in refs. 10,11, which leads to Result 1. Moreover, an improved simulation model is further attained when the $\{|\psi_x\rangle\}_x$ states are confined to an $s$-dimensional subspace, with $s < d$. Details of this model are given in SM. □

Result 1 is versatile due to its generality. For instance, for the simplest case of generic qubit sets, Equation (9) gives $v \leq \frac{1}{2}$. This means that if we shrink the Bloch sphere to half its radius, all quantum sets admit a classical model. Analogous results are implied also for the more complicated state spaces associated with higher dimensions ($d > 2$). Importantly, the visibility bound in Equation (9) is monotonically decreasing with $d$ and tends to zero in the limit of large $d$, implying that it becomes increasingly hard for the classical model to simulate quantum theory when the dimension grows. However, the crucial question now is whether this decreasing power of classical models is fundamental or due to sub-optimality of our specific choice of model. The next result shows that the answer is the former.

**Result 2.** (Classicality of noisy quantum theory) Consider the set of all pure $d$-dimensional quantum states subject to isotropic noise, $\{v|\psi\rangle\langle\psi| + \frac{1-v}{d}\mathbb{1}\}_{|\psi\rangle \in \mathbb{C}^d}$ where $v \in [0, 1]$ is the visibility. A necessary and sufficient condition for classical simulability is given by Equation (9).

**Proof.** The proof consists in showing that the sufficient condition in Equation (9) also is a necessary condition for classical simulability when the set corresponds to all pure states in $d$-dimensional quantum theory. The argument is presented in SM. It builds on symmetries of the set and uses computation techniques from[10,11]. □

Result 2 identifies the precise noise limit at which classical models can simulate quantum theory in dimension $d$. For large $d$, the visibility threshold scales as $v \sim \frac{\gamma - 1 + \log d}{d}$, where $\gamma \approx 0.577$ is the Euler-Mascheroni constant. This quite rapidly approaches zero, thereby attesting to the fading power of classical models for high-dimensional quantum theory. We emphasise that the favourable scaling given by Result 2 only applies to the limit in which the classical model aims to simulate every state allowed in $d$-dimensional quantum theory. For smaller (to be more practically relevant, finite) sets of states, Result 1 only constitutes a lower bound on the power of classical models.

Let us now consider sets that do not correspond to the entire quantum state space. Since Result 1 holds for all sets of the form (8), one should not expect it to be close to optimal when the set only contains a small number of states. We now develop an alternative simulation model that depends explicitly on the number of states appearing in the set. A broad class of sets commonly used in quantum information correspond to the collection of all eigenstates from several different bases. The next result shows how such sets can be classically simulated.

**Result 3.** (Simulation of sets of bases) Consider $M$ different bases of $d$-dimensional Hilbert space and let $\{|\psi_x\rangle\}_{x=1}^{dM}$ be all their eigenstates. After mixing with noise of visibility $v \in [0, 1]$, as in Equation (8), the set is classically simulable if

$$v \leq \frac{1}{M}. \tag{10}$$

Moreover, it is $r$-simulable if $v \leq (r - 1)/(M(d - 1))$.

**Proof.** We provide a sketch of the classical model and refer to SM for details. We build all relevant projectors $\Pi_\lambda$ as those associated with all the $r$-element subsets selected from the $d$ basis-elements in each of the $M$ bases. This gives $M\binom{d}{r}$ projectors, which we index as $\Pi_\lambda \equiv \Pi_{i,j}$, $i = 1, \ldots, \binom{d}{r}$, $j = 1, \ldots, M$, corresponding to preparation devices $\mathcal{P}_\lambda \equiv \mathcal{P}_{i,j}$. We let $\mathcal{P}_{i,j}$ emit the state $|\psi_x\rangle$ if it is an eigenstate of $\mathcal{P}_{i,j}$. Otherwise we let $\mathcal{P}_{i,j}$ emit its maximally mixed state $\frac{1}{r}\Pi_{i,j}$. One can then show that this simulates $\mathcal{E}$ up to the visibility given in (10). □

Result 3 depends on the number of bases, $M$, but not on how they are selected. Therefore, for small values of $M$, this model can significantly improve on Result 1. Comparing them, we see that the classical simulation is enhanced whenever $M \leq \lfloor\frac{d-1}{H_d-1}\rfloor$. This holds for example for $M = 2$ bases in $d = 3$ dimensions.

Our analytical simulation models apply to general sets of pure states with isotropic noise. Apart from the important special case addressed in Result 2, the models are not expected to be optimal. Moreover, many times it is relevant to consider other types of noise than the isotropic case. It is therefore relevant to develop useful numerical methods that can efficiently search for classical models for any given quantum set. We now develop such methods and demonstrate their efficiency.

**Table 2 | Critical visibilities for classical simulation of quantum sets**

| d | N_U | Set | Numerical | Result 1 | Result 3 |
|---|-----|-----|-----------|----------|----------|
| 3 | 3000 | $\mathcal{E}_1$ | 0.6122 | 0.4167 | 0.5000 |
| 3 | 3000 | $\mathcal{E}_2$ | 0.5257 | 0.4167 | 0.3333 |
| 3 | 3000 | $\mathcal{E}_3$ | 0.4567 | 0.4167 | 0.2500 |
| 3 | 20 | $\mathcal{E}_4$ | 0.6722 | 0.4167 | 0.5000 |
| 3 | 100 | $\mathcal{E}_5$ | 0.4290 | 0.4167 | 0.1111 |
| 4 | 1000 | $\mathcal{E}_6$ | 0.3785 | 0.3611 | 0.0625 |

Classical simulation thresholds obtained from Equation (11) for six noisy quantum sets. The sets $\mathcal{E}_1$, $\mathcal{E}_2$ and $\mathcal{E}_3$ correspond to all states in 2, 3 and 4 MUBs, respectively, in dimension $d = 3$. The set $\mathcal{E}_4$ is just two states: $|0\rangle$ and $\frac{|0\rangle + |1\rangle + |2\rangle}{\sqrt{3}}$ in dimension $d = 3$. $\mathcal{E}_5$ and $\mathcal{E}_6$ are the nine and sixteen states forming a SIC-set in dimension $d = 3$ and $d = 4$ respectively. Analytical models based on Results 1 and 3 are included for comparison. $N_U$ is the number of classical devices used in the simulation.

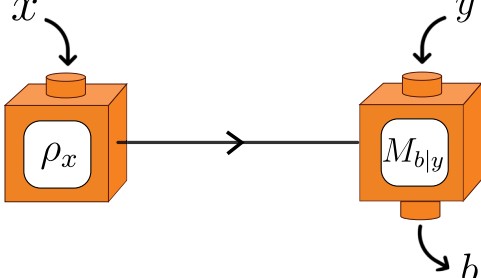

**Fig. 4 | Prepare and measure scenario.** A set of measurements $\{M_{b|y}\}$, where $y$ denotes the measurement choice and $b$ the outcome, is performed on the set $\{\rho_x\}_x$. The outcome statistics is used to test for whether the set defies classical simulation.

To perform a classical simulation, we need a set of preparation devices, each emitting $m$ commuting $d$-dimensional states. Let each preparation device $\mathcal{P}_\lambda$ be identified with a unitary $U$ and the corresponding orthonormal basis $\mathbf{b}_U = \{U|i\rangle\}_{i=1}^d$ of $\mathbb{C}^d$. Thus, the label $\lambda$ is now replaced by the unitary $U$. Since we want the states emitted by each device to commute, we write them in the diagonal form $\tau_{x,U} = \sum_{i=1}^d p(i|x, U)U|i\rangle\langle i|U^\dagger$. Our approach consists in selecting a set of unitaries, $\mathcal{U}$, associating a preparation device with each $U \in \mathcal{U}$ and then using convex programming to find the best classical simulation possible with these devices. In order to also quantify the simulability, we introduce a robustness parameter: if asked to simulate the states $\{\rho_x\}_x$, we search for the largest visibility $v$ for which a classical model exists for the set $\mathcal{E} = \{v\rho_x + \frac{1-v}{d}\mathbb{1}\}_x$. Thus, we obtain a bound on the amount of isotropic noise that must be added to the states in order to find a classical model. This is computed with the following linear program (LP),

$$\max_{v, q, \widetilde{p}} v$$

$$\text{s.t. } v\rho_x + \frac{1-v}{d}\mathbb{1} = \sum_{U \in \mathcal{U}} \widetilde{\tau}_{x,U}, \ \forall x,$$

$$\widetilde{\tau}_{x,U} = \sum_{i=1}^d \widetilde{p}(i|x, U) \, U|i\rangle\langle i|U^\dagger, \ \forall x,$$

$$\widetilde{p}(i|x, U) \geq 0 \ \forall i, x, U, \tag{11}$$

$$\sum_{i=1}^d \widetilde{p}(i|x, U) = q(U), \ \forall x, U,$$

$$\sum_{U \in \mathcal{U}} q(U) = 1, \ q(U) \geq 0,$$

where we have defined unnormalised states $\widetilde{\tau}_{x,U} = q(U)\tau_{x,U}$. Our implementation of this LP is found at[12]. We emphasise that this method can address generic noisy states because $\rho_x$ can be an arbitrary mixed state; the isotropic noise appearing in (11) serves only as a quantifier of the simulability of $\{\rho_x\}_x$.

The efficiency of the method depends strongly on the selection of the set of unitaries, $\mathcal{U}$. In SM, we discuss three different ways of making this selection and compare their relative performance. There, we also show how this method with small modifications can also be extended to search for classical models with complexity $r < d$. We demonstrate the relevance of this method by showing how it surpasses the analytical models for several natural choices of sets. In Table 2 we have considered six standard types of sets: three based on mutually unbiased bases (MUBs)[13], two based on symmetric informationally complete (SIC) sets[14] and one minimal set consisting of just two states with maximal relative coherence. In all cases, we select a large number

of unitaries, $\mathcal{U}$, associate a classical state-preparation device with each unitary, and improve on our best available analytical method.

### Detecting absolute quantum coherence

The models developed in Section "Results" "Simulation of quantum sets" provide sufficient conditions for classicality. In this section, we address the complementary question, namely that of finding necessary conditions. The violation of such conditions implies that no classical model exists, which can be regarded as a certificate of absolute quantum coherence. When falsifying the existence of classical models, it is relevant to do so in way that is compatible with experimental tests, so that these features can be harnessed. Therefore, we systematically develop criteria that can be measured in a prepare-and-measure scenario based on well-calibrated measurement devices. Using well-calibrated devices is a natural choice since we aim to witness properties of the set of states emitted by Alice.

Consider that an uncharacterised preparation device emits a set of states $\mathcal{E} = \{\rho_x\}_{x=1}^m$. Instead of tomographically reconstructing the set, which is well-known to be an expensive procedure, we instead aim to infer its quantum properties only by performing a few measurements; see Fig 4. We let the experimenter select a number of measurements to perform, $\{M_{b|y}\}$, where $y$ denotes the measurement choice and $b$ the outcome. Via the Born rule, this is associated to the probability distribution $p(b|x, y) = \text{tr}(\rho_x M_{b|y})$. Since the classical set $\mathcal{S}$ is a convex, every set $\mathcal{E} \notin \mathcal{S}$ can be detected as such through a separating hyperplane. Therefore, we can without loss of generality consider linear witness-type inequalities of the form

$$W(\mathcal{E}) \equiv \sum_{b,x,y} c_{bxy}\text{tr}(\rho_x M_{b|y}) \leq \beta^C, \tag{12}$$

where $c_{bxy}$ are some real coefficients and $\beta^C$ is a tight bound satisfied by all classical models. Thus, a violation of this inequality implies the failure of all classical models. This approach may be viewed as an analogy to entanglement witnessing[15] since it (in contrast to most literature on prepare-and-measure scenarios) does not adopt a black-box description of the measurement device. Note that this approach also can be used to bound the classical simulation complexity, $r$ by finding bounds, $\beta_r^C$, on $W$ respected by all $r$-simulable classical sets instead.

The central question is how to compute upper bounds on $\beta^C$ for a witness with arbitrary coefficients $\{c_{bxy}\}$ and arbitrary choice of measurements $\{M_{b|y}\}$. The next result shows how this problem can be addressed.

**Result 4.** (Witness method) Consider a witness of the form in Equation (12), where $\{M_{b|y}\}$ is a set of measurements and $\{c_{bxy}\}$ are real

coefficients. All classical models satisfy

$$\beta^C \equiv \max_{\mathcal{E} \in \mathcal{S}} W(\mathcal{E}) = \max_\gamma g_\gamma, \tag{13}$$

where

$$g_\gamma \equiv \max_{\{\varphi_i\}} \sum_{i=1}^d \sum_x D_\gamma(i|x) \langle \varphi_i | \mathcal{O}_x | \varphi_i \rangle. \tag{14}$$

Here, the maximisation is over all orthonormal bases $\{|\varphi_i\rangle\}_{i=1}^d$. The list $\{D_\gamma(i|x)\}_\gamma$ are all deterministic input-output strategies and $\mathcal{O}_x = \sum_{b,y} c_{bxy} M_{b|y}$.

**Proof.** Assume that $\mathcal{E}$ has a classical model, i.e. it admits the form (2). Since the witness is linear and $\mathcal{S}$ is convex, $W$ attains its maximal value for a deterministic strategy. We can therefore restrict to the set $\{\tau_x\}_x = \{\tau_{x,\lambda'}\}_x$ emitted from a single preparation device $\mathcal{P}_{\lambda'}$. Since it satisfies $[\tau_x, \tau_{x'}] = 0$, we can write it in a diagonal basis $\tau_x = \sum_{i=1}^d p(i|x)|\varphi_i\rangle\langle\varphi_i|$ where $\{|\varphi_i\rangle\}_i$ are orthonormal. Then, we have $W(\mathcal{E}) \leq \sum_{i=1}^d \sum_x p(i|x)\langle\varphi_i|\mathcal{O}_x|\varphi_i\rangle$. Next, we decompose each conditional probability as a convex combination of all deterministic distributions, i.e. $p(i|x) = \sum_\gamma p(\gamma) D_\gamma(i|x)$ where $D_\gamma(i|x) \in \{0, 1\}$. Since the optimal value of $W$ must occur for a deterministic choice of $p(\gamma)$, we get the final result $\beta^C = \max_\gamma \max_{\{\varphi_i\}} \sum_{i=1}^d \sum_x D_\gamma(i|x)\langle\varphi_i|\mathcal{O}_x|\varphi_i\rangle$. □

Equipped with Result 4, the problem of finding a bound on the witness (12) respected by all classical models is reduced to computing upper bounds on $g_\gamma$. We do this by relaxing the optimisation problem in Equation (14) to a semidefinite program (SDP)[16], which can then be evaluated efficiently using standard techniques. To this end, we relax the orthonormal basis constraint on $\{|\varphi_i\rangle\}_i$ to a quantum measurement $\{E_i\}_i$ where all outcome operators have unit trace. The SDP reads

$$g_\gamma^\uparrow \equiv \max_{\{E_i\}} \sum_{i=1}^d \sum_x D_\gamma(i|x) \mathrm{tr}(\mathcal{O}_x E_i)$$
$$\text{s.t. } E_i \succcurlyeq 0, \ \mathrm{tr}(E_i) = 1, \ \sum_i E_i = \mathbb{1} \ \forall i, \tag{15}$$

and $g_\gamma^\uparrow \geq g_\gamma$ holds by construction. This can be evaluated for every $\gamma$. We can then obtain our desired bound in Equation (12) selecting the largest value, namely $\beta^C \leq \max_\gamma \{g_\gamma^\uparrow\}$. Our implementation of this program is available at[17].

We demonstrate the usefulness of this method through relevant case-studies. Let us choose $d = 3$ and select our measurements $\{M_{b|y}\}_{b=1,y=1}^{3,N}$ as corresponding to $N = 2, 3, 4$ MUBs. These bases have the defining property that if $|\phi_1\rangle$ and $|\phi_2\rangle$ are any pair of elements from any two distinct bases, their overlaps are uniform, i.e. $|\langle\phi_1|\phi_2\rangle|^2 = \frac{1}{d}$. Now we must select the coefficients $c_{bxy}$ in our witness. For this selection, we consider sets of size $|\mathcal{E}| = 3N$ and we index the states as $\rho_x = \rho_{j,k}$ where $x = (j, k)$ for $k \in \{1, 2, 3\}$ and $j \in \{1, ..., N\}$. The target set, whose absolute coherence we want to reveal, is comprised of all three states in all $N$ bases. Therefore, if we prepare the state $(j, k)$ and then measure basis $j$, we expect to obtain the outcome $b = k$. We therefore select the witness as the sum of the probability of these events, i.e. $c_{bxy} = c_{bjky} = \delta_{b,k}\delta_{j,y}$. The witness becomes

$$W_N(\mathcal{E}) = \sum_{k=1,2,3} \sum_{j=1}^N \mathrm{tr}(\rho_{j,k} M_{k|j}). \tag{16}$$

We use Result 4 and the SDP relaxation in Equation (15) to compute bounds on $W_N$ for all classical models of $\mathcal{E}$. The bounds are

$$W_2 \leq 4.6667, \ W_3 \leq 6.4115, \ W_4 \leq 7.7835. \tag{17}$$

These are resoundly violated by our target set of $N$ MUBs, which by construction achieves $W_N = 3N$. Importantly, due to the large violations, these witnesses can also detect absolute coherence in many other sets $\mathcal{E}$. For instance, if we consider the mixture of our target set with noise, as in Equation (8), we find a violation for

$$v \gtrsim 0.6667, \ v \gtrsim 0.5686, \ v \gtrsim 0.4729, \tag{18}$$

respectively. This shows that by including more bases, we obtain increasingly noise-robust certificates of quantum properties. Note also that these sets are those respectively labelled $\mathcal{E}_1$, $\mathcal{E}_2$ and $\mathcal{E}_3$ in Table 2. Combining the results from (18) and the table, we see that there is only a small range of $v$ left in which we have not determined whether or not the set is classically simulable.

## Foundational aspects

It is interesting and potentially also practically useful to ask whether the proposed notion of set classicality admits connections to established foundational concepts in quantum theory. In this section, we address this question. We first show that tests of Einstein-Podolsky-Rosen steering can be transformed into witnesses testing classical simulability. Then, we show that set classicality implies a special form of joint measurability. These connections also allow us to export tools from steering and joint measurability into the analysis of classical simulability for quantum states.

Steering is the possibility to remotely influence the state of a particle by performing local measurements on another particle with which the former is entangled. Denoting the state of the two particles by $\rho_{AB}$ and the local measurements by $\{A_{a|x}\}$, where $x$ is the input and $a$ is the output, the remote states of the other particle and their conditional probabilities are characterised by the assemblage $\sigma_{a|x} = \mathrm{tr}_A(A_{a|x} \otimes \mathbb{1}\rho_{AB})$. If the set $\{\sigma_{a|x}\}$ cannot be modelled by means of a local hidden variable, it is said to be steerable[10]. Much research effort has been invested in studying steering with a pair of entangled qubits[5]. To reveal the steering, one commonly lets Alice perform dichotomic measurements on her qubit. These are associated with obseravables $A_x \equiv A_{0|x} - A_{1|x}$. Bob performs suitable rank-one projective measurements on his qubit, corresponding to the observable $B_y$. The standard approach is to consider a steering inequality of the form

$$\sum_{x,y} s_{x,y} \langle A_x, B_y \rangle_{\rho_{AB}} \leq \zeta, \tag{19}$$

for some arbitrary real coefficients $s_{x,y}$, where the tight bound $\zeta$ holds for all local-variable models but can be violated by quantum theory. Our next result shows that any such steering inequality can be transformed into a witness-type criterion, as in section II F, respected by all classical set models.

**Result 5.** (Steering equivalence for qubits) Consider a two-qubit steering inequality of the form (19). Then the inequality

$$W(\mathcal{E}) \equiv \sum_{x,y} s_{x,y} \mathrm{tr}(\rho_x B_y) \leq \zeta, \tag{20}$$

is tight and holds for all classical sets $\mathcal{E} = \{\rho_x\}_x$.

**Proof.** The proof is given in SM. The main idea is to use Result 4 to express the classical bound of $W(\mathcal{E})$ as the set of optimisation problems associated with $\{g_\gamma\}$, and then show that this set of optimisation problems is identical to the characterisation of the bound in the steering inequality (19). □

A direct consequence of this result is that results known for two-qubit steering can be re-interpreted to characterise the classicality of

qubit sets. For instance, ref. [18] identifies many steering inequalities of the form (19) that display both strong noise tolerance and strong robustness to particle-loss[19]. Via Result 5, we obtain the corresponding witnesses for falsifying classical simulation of quantum sets. Importantly, these inequalities will inheret the favourable noise-tolerance properties of the original steering inequality. In a similar vein, Ref. [20] identified the optimal set of $m$ measurements for Alice to optimise the noise tolerance of a steering test on a maximally entangled state. By Result 5, these optimal measurement selections can be re-interpreted as the set of $m$ single-qubit states that can tolerate the most isotropic noise before admitting a classical model.

We now go a step further and show that, in general, set classicality implies a specific form of joint measurability. A set of measurements $\{M_{a|x}\}$ is said to be jointly measurable if all the separate measurements (indexed by $x$) can be obtained as post-processings of the outcome of just a single measurement[4]. Thus, $\{M_{a|x}\}$ is jointly measurable if it can be expressed as $M_{a|x} = \sum_\mu p(a|x, \mu)G_\mu$ for some measurement $\{G_\mu\}$. To show how this can be related to classical sets, we must first associate a set of measurements to our set $\mathcal{E} = \{\rho_x\}_x$. To this end, it is convenient to define an extended set $\mathcal{E}' = \mathcal{E} \cup \{\frac{1-\rho_x}{d-1}\}_x$. Evidently, by simply discarding the second set, $\mathcal{E}' \in \mathcal{S}$ implies $\mathcal{E} \in \mathcal{S}$. What is less obvious is that also the converse holds (see SM) and hence $\mathcal{E} \in \mathcal{S}$ if and only if $\mathcal{E}' \in \mathcal{S}$. Thus, we can now study $\mathcal{E}'$ instead of $\mathcal{E}$. It has the advantage that it can naturally be associated with a set of binary measurements $\mathcal{M} = \{M_{0|x}, M_{1|x}\}_x$ where $M_{0|x} = \rho_x$ and $M_{1|x} = \mathbb{1} - \rho_x$. Now, we show that the classicality of the set implies the joint measurability of the binary measurement set.

**Result 6**. (Connection to joint measurability) Let $\mathcal{E} = \{\rho_x\}_x$ be a $d$-dimensional set and let $\mathcal{M} = \{\rho_x, \mathbb{1} - \rho_x\}_x$ be an associated set of binary measurements. Then it holds that

$$\mathcal{E} \text{ has a classical model } \Rightarrow \mathcal{M} \text{ is jointly measurable}. \qquad (21)$$

Moreover, for the case of qubits ($d = 2$), set classicality and joint measurability are equivalent.

**Proof**. The classicality of $\mathcal{E} = \{\rho_x\}_x$ is equivalent to the classicality of $\mathcal{E}' = \mathcal{E} \cup \{\frac{1-\rho_x}{d-1}\}_x$. Hence there exists a probability density function $q(\lambda)$ and a set of states $\{\tau_{x,\lambda}\}$ so that Equation (2) holds. Due to commutation, we have $\tau_{x,\lambda} = \sum_{i=1}^d P(i|x,\lambda)\Phi_{i,\lambda}$ for some basis $\{\Phi_{i,\lambda} = |\phi_{i,\lambda}\rangle\langle\phi_{i,\lambda}|\}_{i=1}^d$. We build the parent measurement for $\mathcal{M}$ by $G_\mu \equiv G_{i,\lambda} = q(\lambda)\Phi_{i,\lambda}$, while the probability distribution $p(a|x, \mu) \equiv p(a|x, i, \lambda)$ is given by $p(0|x, i, \lambda) = P(i|x, \lambda)$ and $p(1|x, i, \lambda) = 1 − P(i|x, \lambda)$. It is straightforward to check that with these definitions the measurements $\mathcal{M}$ are jointly measurable. To see the equivalence for qubits, we notice that a given parent measurement for $\mathcal{M}$ can be used to build the classical simulation of $\mathcal{E}'$ and therefore also of $\mathcal{E}$. We refer to SM for details. □

For qubit measurements of the above type, necessary and sufficient conditions for joint measurability are known for two[21], three[22,23] and an arbitrary number of measurements[24]. This implies a corresponding characterisation of classical sets of qubit states. However, when $d > 2$ the relevant notion of joint measurability of binarised measurements is more unconventional, since one of the two outcomes is associated with a unit-trace outcome operator. In light of this, it is natural to ask whether set classicality and joint measurability of binarisations are equivalent concepts in all dimensions, i.e. whether equivalence in Result 6 also holds for $d > 2$. We now answer this in the negative through a counter-example based on qutrits ($d = 3$). For this, we revisit the witness (16) and the set of six states corresponding to the eigenstates of $N = 2$ MUBs. Constructing the set of six corresponding binary measurements $\mathcal{M}$, it is known to be jointly measurable only when the states have a visibility of at most $v \le \frac{1}{2}\left(1 + \frac{1}{1+\sqrt{3}}\right) \approx 0.68$[25]. In contrast,

our discussion in section II F showed that this set cannot be classically simulated when $v \gtrsim 2/3$; see Equation (18). Hence, for $2/3 < v \lesssim 0.68$ the set is not classically simulable but $\mathcal{M}$ is still jointly measurable.

## Discussion

Classicality has many different meanings in quantum information. In this work, we have put forward a notion of classicality that pertains to quantum state-preparation devices without a priviledged basis. It is a framework distinct from other manifestations of classical properties, such as those associated with Bell inequality tests, contextuality or Wigner quasi-probability distributions. Our framework draws inspiration from the basic textbook knowledge that commuting sets of quantum states can be jointly diagonalised and therefore feature no relational superpositions. Unless one insists on a priviledged basis, this renders such states effectively classical. Here, our classical models impose joint diagonalisability on state preparation devices instead of on the density matrices. This is motivated by an operational picture in which the absense of superposition is implemented on actual devices rather than on associated Hilbert space concepts. By operating only devices of this type, and permitting them to interact with each other using standard pre- and post-processing resources, we have shown that the capability of classical resources is significantly greater than what is captured by commutation.

The results of this article are

(I) explicit classical simulations of quantum states,
(II) methods for ruling out the existence of classical models for quantum states,
(III) relations between classical set models and established fundamental concepts in quantum theory.

Results of type (I) do not only shine light on the power of classical models for quantum theory but they are also relevant from a quantum technology perspective. For example, as we discussed in section II B, classical models prevent quantum correlations in the semi-device-independent prepare-and-measure scenario. Such scenarios are often used for quantum cryptography protocols. With enough noise and/or loss in such protocols, the states effectively seen by Bob will admit a classical model. Such a model can be viewed as an explicit hacking attack deployed by an adversary who can now listen in on the communication without disturbing the states.

Results of type (II) make possible the certification of absolute coherence in sets of quantum states, i.e. that devices must be able to generate relational superpositions. Since this is an important benchmark for quantum preparation devices, the methods that we have introduced are designed to be versatile for experiments: they apply to general quantum sets, they are significantly robust to noise and/or imperfections, and they can be adapted to the measurements handy in any given physical system or setup. We note the open problem of finding more computationally efficient methods for these criteria.

Furthermore, results of type (III) are relevant for contextualising the role of our proposed notion of set classicality among established foundational concepts in quantum theory. As we have shown, this also leads to useful tools for hands-on analysis of quantum versus classical sets of states. For example, we have shown that results from the substantial body of literature on both measurement incompatibility and Einstein-Podolsky-Rosen steering can be adapted to conduct tests of quantum state sets.

The problem of certifying quantum properties in a set of states has been extensively studied in the literature, but mainly from the point of view of device-independent quantum information[7–9]. Typically, this black-box picture causes the gap between quantum and classical correlations to diminish when the dimension grows[26–28]. Our type (II) results may be viewed as the analogous problem but treated in the framework of standard quantum information (i.e. no device-independence) in which auxiliary devices used to probe the states are assumed

to be controlled. This is reminiscent of the distinction between Bell nonlocality and entanglement. Our results show that, in contrast to the device-independent picture, classical models for sets of quantum states can become increasingly weak as the dimension grows. In particular, we identified the precise noise rates required for classically simulating all states allowed in $d$-dimensional quantum theory and found that it tends to unit. On the one hand, this provides a foundational justification for recent years' optimism in using high-dimensional quantum systems to overcome practical obstacles in quantum information. On the other hand, as our witness-based and numerics-based results have shown, equally favourable scaling cannot be expected when classical models aim to simulate only a more restricted set of states. The latter is important because practical quantum information protocols are based on a specific number of particular states. Nevertheless, our results have exemplified how using only a modest number of well-selected states one can significantly weaken classical simulability. This paves a way to harnessing the advantages of higher dimension for practically relevant quantum devices.

## Data availability
All data are available in the main text or the supplementary materials.

## Code availability
Our implementation of the optimization problems is available at https://doi.org/10.5281/zenodo.17474952. and https://doi.org/10.5281/zenodo.17473753.

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

## Acknowledgements
A.B and M.J.R acknowledge the hospitality of the Lund quantum information group. We thank Roope Uola and the Geneva quantum correlations group for discussions. This work is financially supported by the Swedish Foundation for Strategic Research, by the Krapperup Foundation, by the Wenner-Gren Foundations, by the Knut and Alice Wallenberg Foundation through the Wallenberg Center for Quantum Technology (WACQT) and by the Swedish Research Council under Contract No. 2023-03498. The work of A.B. is supported through the FPI grant PRE2020-095867 funded by MCIN/AEI/10.13039/501100011033. M.J.R. acknowledges financial support by the Vienna Doctoral School in Physics (VDSP).

## Author contributions
G.C., A.B., M.J.R., and A.T. participated in the development of the theory and in writing the manuscript.

## Funding

## Competing interests
The authors declare no competing interests.
