## [Transparent Peer Review file · Nature Communications]

Operationally classical simulation of quantum states

Corresponding Author: Dr Armin Tavakoli

Version 0:

Reviewer comments:

Reviewer #1

(Remarks to the Author)
See the attached report.

[Editorial Note: see end of file]

(Remarks on code availability)

Reviewer #3

(Remarks to the Author)

This paper develops a new “operationally classical” framework for modeling quantum state sets, based on the idea that each classical state-preparation device can only output commuting states in some basis (with the basis allowed to differ across devices), while a random variable stochastically chooses which device to use. This extends the standard commutation-based definition of classicality and allows for classical simulation of many quantum sets that do not commute. The authors present several main results: (i) Exact thresholds for isotropic noise above which the entire d -dimensional quantum state space becomes classically simulable, showing that the critical visibility decreases with dimension and scales as $1 - \log(d)/d$ for large d ; (ii) Analytical bounds for simulating noisy sets of pure states and sets built from multiple orthonormal bases, as well as a measure of “simulation complexity” based on the subspace dimension used by each device; (iii) Numerical methods using convex optimization to find classical models for arbitrary noisy sets, outperforming the analytical bounds in many examples; (iv) Experimental witnesses—linear inequalities—capable of certifying that no classical model exists; and (v) Foundational connections showing that the framework implies a form of joint measurability, and that qubit set classicality is equivalent to certain Einstein–Podolsky–Rosen steering conditions.

Overall, the work maps the boundary between classical and quantum state-sets in a basis-independent, operational manner, offers tools for both constructing and ruling out classical simulations, and highlights how the power of classical models declines with dimension. The paper is well written, and the results are presented with sufficient clarity to follow the arguments. The work is original, and the proofs are rigorous and technically sound. I really enjoyed reading the paper.

While I find the topic interesting and worthy of publication in some form, I am not convinced about the general significance of the results in the limit of large dimensions. As the authors themselves show, increasing the system size makes the vast majority of states fail to be classically simulable—a conclusion that might be anticipated given that the classically simulable sets are parametrized by fewer variables than the full Hilbert space, bearing similarities to entanglement vs. separability. At the same time, current experiments routinely demonstrate coherent control over hundreds of qubits or qudits with large d , already operating in a regime where, under the paper’s definition, almost all states would not be classically simulable. This raises the question of whether the main results retain their practical significance in such regimes. The authors remark in the discussion that “This shows that the advantages of high dimensionality have the potential to become very significant.” My concrete question, then, is whether and in what way the methods developed here could help to harness this potential.

(Remarks on code availability)

Version 1:

Reviewer comments:

Reviewer #1

(Remarks to the Author)

Please See the attached report.

[Editorial Note: see end of file]

(Remarks on code availability)

Reviewer #3

(Remarks to the Author)

The revised manuscript and the response of the authors clarifies my concerns about the applicability of the techniques for large systems. My main concern was not the number of hidden parameters λ , but the fact that the fraction of stimuable states seems to approach zero in the limit of large N . However, as the authors correctly point out, and now also highlight more in the revised manuscript, this does not apply to smaller sets of (physically) relevant states, for which the authors provide examples of the applicability of their results. This resolves my concern and I thank the authors for their response. I am happy to recommend acceptance to Nature Communications.

(Remarks on code availability)

Reviewer 1

The manuscript presents technically correct results and introduces a new definition of classicality. Nonetheless, we find physical justification for adopting this particular definition missing. While the work has merit and could warrant publication, significant conceptual clarifications are needed. Our specific comments are:

We thank the Reviewer for their reading of our manuscript. Below, we reply to their comments point by point.

- *The authors claim that the set S of classical models is convex: if ρ_x in S and σ_x in S , then $p\rho_x + (1-p)\sigma_x \in S$ for any $p \in [0, 1]$. This indeed follows. However, does the convex hull of a classical set necessarily remain classical? If yes, a proof should be provided; if not, a counterexample is needed.*

In the resubmission, we clarify that the convex hull of the classical set is identical to the classical set, i.e. the set obtained from $\{p\rho_x + (1-p)\sigma_y\}_{x,y}$ is classical for any ρ_x in S and σ_y in S . The proof is given in a newly added section of supplementary material.

- *The manuscript does not convincingly argue why the proposed construction should be regarded as a definition of classicality. A natural operational interpretation arises in the PAM scenario: λ acts as a shared random variable; Alice, given x and λ , prepares a diagonal state labeled by z , sends z via a d -level classical system, and Bob reconstructs the state. On average, this yields ρ_x . This reasoning would justify the definition, yet the authors seem not to adopt it.*

For instance, the set $0, +$ can be classically prepared by Alice sending one classical bit, yet this set is not classical by their definition (or since the states do not commute). Moreover, it never generates nonclassical PAM statistics. Thus, their definition diverges from the operational notion of classicality. We urge the authors to clarify what operational principle their framework captures.

Indeed, our definition was inspired by the PAM scenario and we agree with the referee that its operational relevance is strengthened by making that connection explicit. In the revision, we have added a new subsection (IIB + new figure) in which we discuss the relation of our definition to the PAM scenario and we also add a highlight of that in the abstract. We thank him/her for this useful suggestion.

Our definition of classicality for a set of states is to the PAM scenario what separability vs entanglement is to the Bell scenario. In other words, in analogy with how entanglement tells us whether a bipartite quantum state is classical (separable) or not, our definition tells us whether a set of quantum preparations is classical or not. Just like entanglement being necessary for nonclassical statistics in the black-box Bell scenario (nonlocality), our definition is necessary for nonclassical statistics in the black-box PAM scenario. Neither entanglement nor our definition is sufficient for the respective black-box statistics advantage: this is the reason why the states $|0\rangle$ and $|+\rangle$ represent a non-classical quantum set but cannot produce nonclassical statistics in the black-box PAM scenario. $|0\rangle$ and $|+\rangle$ are in a relational superposition (not classical) but if we

have no knowledge of the measurement performed on them, the quantum feature is washed out in the black-box statistics. This is now clarified in the new subsection.

In further analogy, the operational principle behind entanglement is that it cannot be generated with local quantum devices, even if they may communicate classically. The operational principle behind our definition is that the set of preparations cannot be generated by devices that are unable to produce superposition states, even if they may be classically pre- and post-processed. The “no-superposition” principle is at the heart of what makes a preparation device classical and our definition captures the strongest possible model that respects this limitation.

Finally, thanks to the Referee’s comments, we also noted a nice feature regarding our example of BB84 states (in section IIC). In this example, our classical model is both necessary and sufficient for a statistics advantage in the black-box PAM scenario. It implies that there is no better protocol than the quantum random access code for revealing the nonclassicality of the BB84 set. We have added this comment to the manuscript to further showcase the usefulness of our definition.

- *Furthermore, their construction of witnesses depends explicitly on Bob’s measurement choice, unlike conventional semi-device-independent PAM witnesses such as RAC success probabilities. This suggests that the proposed framework constitutes a new paradigm where measurement settings are fixed by the witness itself. If so, the authors should clearly articulate the operational meaning of nonclassicality in this setting.*

The referee is right that what we do is different from conventional semi-device-independent PAM witnesses and the reason is directly connected to the previous point.

Our definition concerns the classicality of the set of states themselves and not the statistics they generate in a black-box PAM scenario. Therefore, we use the PAM scenario but with a fixed measurements for Bob to witness the non-classicality of the set of states. Continuing the analogy with entanglement, it is well-known that if we have some entangled state we can witness it by making specific measurements on its parts. If we had assumed that these measurements are unknown, we would have called this a Bell inequality test. Similarly, we witness the non-classicality of a set of quantum states by making specific measurements on it. If we had assumed that these measurements are unknown, we would have called it a semi-device-independent PAM test. Thus, if in either of these scenarios we had adopted black-box measurements, then we would no longer be testing the classical model (separability/entanglement or our definition, respectively) for the quantum state(s) but the classical model (local hidden variable, or semi-device-independent PAM respectively) for the statistics. While tests of whether semi-device-independent PAM statistics is classical is long-

established but our work is the first to define what it means for the underlying states to admit a classical model.

Our general revision conveys these clarifications, including in the beginning of the relevant section (IIF) and under Eq. 12.

- *Several section references in the Supplementary Information appear incorrect and should be carefully checked.*

In the resubmission, we have fixed this.

- *Suggestion:*

It would be valuable to map their definition onto an operational task, analogous to how Bell inequalities certify entanglement or steering inequalities certify one-sided entanglement. Establishing such a task would give strong justification for calling these models "classical".

As mentioned above, we fully agree with the Referee's suggestion and we have now included the connection to PAM scenarios to convey this task-based relevance. Please see section IIB and the new Fig. 2. Also, our previously mentioned example of generating non-classical PAM statistics from the BB84 states is a newly added example of an operational task fully solved thanks to our definition.

- *There is a long tradition of defining classicality through Wigner-type distribution functions [Annals of Physics 176, 1-21 (1987); J. Math. Phys. 47, 122107 (2006); J. Phys. A: Math. Gen. 39 1405 (2006) etc], a notion that also enjoys operational justification from computational and communication perspectives [Phys. Rev. Lett. 109, 230503 (2012); Nature 510, 351–355 (2014); arXiv.2506.19369 etc]. Unfortunately, the present manuscript does not discuss how the newly proposed definition offers a stronger or more compelling justification compared to this well-established approach.*

We thank the Referee for their comment. We aware the literature on Wigner distributions and will now carefully clarify why we did not comment on it.

It is well-known that the term "classicality" is in quantum information used in many different ways. For contextuality, it means a non-contextual hidden variable model. For nonlocality, it means local hidden variable model. For bipartite states, it means separability. As noted before by the referee, in semi-device-independent PAM scenarios it has yet another meaning. The list can be continued, etc. However, these different forms of classicality often cannot be compared with each other since they concern different aspects of classical vs quantum physics. This is the reason we did not comment on Wigner-type nonclassicality: it is another, very different, form of nonclassicality (a manifestation of the uncertainty principle in phase-space). In the same way that nonclassicality in Bell scenarios or semi-device-independent PAM scenarios do not have any

strong relations to Wigner-type nonclassicality, also our definition of classicality of preparation devices also does not have any immediately meaningful connection to it.

For additional clarity, we give a few examples of how our notion of nonclassicality for preparation devices cannot meaningfully be compared to Wigner-type nonclassicality.

Firstly, Wigner-type nonclassicality is possible for a single state (since it exhibits uncertainty in position and momentum). This is not possible for our notion since it is defined for sets of states (a single-state preparation device is trivially classical in our work). Secondly, as cited by the referee, Wigner-type classicality has been linked to efficient simulation of quantum circuits on a classical computer. This is because of its correspondence to uncertain relations. That connection will therefore not be possible for our notion: what is efficiently simulable on a classical computer is a different perspective than what is efficiently preparable without superpositions (our notion). Thirdly, it is well-known that stabilizer states are Wigner-type classical (efficiently simulable on a classical computer), but they are necessarily non-classical when it comes to simulating a preparation device that outputs them (our notion). After all, they are at the heart of the BB84-type protocols.

To avoid this pit-fall, we now start the "Discussion" section with clarifying that our notion of classicality is distinct and independent from the several other notions of classicality commonly used in quantum information.

In conclusion, while the results are technically correct, the manuscript in its current form does not provide a sufficiently strong conceptual foundation to warrant publication in Nature Communications.

We thank the Referee for their comments on our manuscript. We hope he/she appreciates our replies and revisions and that they will be willing to reconsider their recommendation.

Reviewer 2

This paper develops a new "operationally classical" framework for modeling quantum state sets, based on the idea that each classical state-preparation device can only output commuting states in some basis (with the basis allowed to differ across devices), while a random variable stochastically chooses which device to use. This extends the standard commutation-based definition of classicality and allows for classical simulation of many quantum sets that do not commute. The authors present several main results: (i) Exact thresholds for isotropic noise above which the entire d -dimensional quantum state space becomes classically simulable, showing that the critical visibility decreases with dimension and scales as $1 - \log(d)/d$ for large d ; (ii) Analytical bounds for simulating noisy sets of pure states and sets built from multiple

orthonormal bases, as well as a measure of “simulation complexity” based on the subspace dimension used by each device; (iii) Numerical methods using convex optimization to find classical models for arbitrary noisy sets, outperforming the analytical bounds in many examples; (iv) Experimental witnesses—linear inequalities—capable of certifying that no classical model exists; and (v) Foundational connections showing that the framework implies a form of joint measurability, and that qubit set classicality is equivalent to certain Einstein–Podolsky–Rosen steering conditions.

Overall, the work maps the boundary between classical and quantum state-sets in a basis-independent, operational manner, offers tools for both constructing and ruling out classical simulations, and highlights how the power of classical models declines with dimension. The paper is well written, and the results are presented with sufficient clarity to follow the arguments. The work is original, and the proofs are rigorous and technically sound. I really enjoyed reading the paper.

We thank the Reviewer for their careful reading of our manuscript and for the appreciation of our work. We are very happy to hear that they really enjoyed reading our paper! Below, we reply to their questions.

While I find the topic interesting and worthy of publication in some form, I am not convinced about the general significance of the results in the limit of large dimensions. As the authors themselves show, increasing the system size makes the vast majority of states fail to be classically simulable—a conclusion that might be anticipated given that the classically simulable sets are parametrized by fewer variables than the full Hilbert space, bearing similarities to entanglement vs. separability.

We believe this is a shaky motivation for anticipating our conclusion.

The number of variables that parameterize the classical model is infinite because the model can in principle use (uncountably) infinitely many inequivalent classical devices (labeled by λ). A related example is that of comparing bipartite Bell nonlocality to local hidden variable models. The latter can also have infinitely many variables and after decades of work on the topic there is still no example in which nonlocality exhibits a scalable gap to local hidden variable models. In contrast, we proved that a scalable gap is possible in our scenario. We find it hard to anticipate that based on variable counting.

At the same time, current experiments routinely demonstrate coherent control over hundreds of qubits or qudits with large d , already operating in a regime where, under the paper’s definition, almost all states would not be classically simulable. This raises the question of whether the main results retain their practical significance in such regimes.

The Referee raises an important question whose answer we wish to clarify.

The classical models are defined for sets of quantum states. In Result 2, we consider a hypothetical quantum device that can prepare every single state in d -dimensional quantum theory. In this “full state space” limit, we indeed prove that for large d the visibility of classical simulation tends to zero for large dimension.

However, it is essential to appreciate that this result only holds in this conceptual limit where the quantum device actually prepares *every* state allowed in the theory: it shows us the potential that high dimension could unlock. However, a real quantum preparation device does of course only prepare some specific (finite) set of states, e.g. those used in the BB84 protocol or similar. Classically simulating such a device is far easier. For example, simulating all qubit states requires visibility 50% whereas simulating the four BB84 states requires 71%. Therefore, it is not the case that more or less any large-scale device will break classical limitations. The methods developed in our paper can assess whether the states generated from specific devices (e.g. BB84) admit a classical model.

The authors remark in the discussion that “This shows that the advantages of high dimensionality have the potential to become very significant.” My concrete question, then, is whether and in what way the methods developed here could help to harness this potential.

As mentioned above, the ‘potential’ comes from Result 2 showing that classical models increasingly fail to simulate the full state space of quantum theory for large dimension. To harness this potential, one must show that real-life devices, limited to only specific and finitely many states, are good enough to benefit from the “would be” advantages of the full state space.

In our manuscript, sections IIE and IIF are respectively dedicated to finding classical models and falsifying classical models for such specific sets of preparations. The methods in section IIF are tailored for practical implementation (which we now point out in the revision).

For example, in section IIF we consider a commonly used set of states in quantum information: those that form $N=2,3,4,\dots$ mutually unbiased bases in dimension d . In eq18 we consider qutrits and prove that while simulating the full state space would put the visibility as low as 42%, we can start tapping into this power using just a small number of states. With 6 states we can reach 67%. With 9 states we reach 57%. With 12 states we reach 47%. This clearly indicates that a reasonable number of well-selected states can harness the power of high-dimensional quantum states. These results are made possible thanks to the methods developed in our manuscript.

In our revised Discussion section, we highlight the above discussion and how our work paves the way for concrete advantages from real-life devices.

Reply to Referee 1

We thank the Referee for their second report on our manuscript.

They have provided the following new comment.

Authors may consider adding a short “Outlook” paragraph in the Discussion suggesting future experimental implementations of the proposed witnesses.

We believe our current Discussion section adequately covers also the outlook aspects of our results. Therefore we have left it unchanged.

Reply to Referee 3

The revised manuscript and the response of the authors clarifies my concerns about the applicability of the techniques for large systems. My main concern was not the number of hidden parameters λ , but the fact that the fraction of stimuable states seems to approach zero in the limit of large N . However, as the authors correctly point out, and now also highlight more in the revised manuscript, this does not apply to smaller sets of (physically) relevant states, for which the authors provide examples of the applicability of their results. This resolves my concern and I thank the authors for their response. I am happy to recommend acceptance to Nature Communications.

We thank the Referee for their second report on our manuscript. We are glad that they appreciate our replies and revisions, and that they recommend publication.

Review Report: NCOMMS-25-52887-T

Summary of the results:

The authors introduce a generalized notion of operationally classical states. Traditionally, a set of states is regarded as classical if they can all be simultaneously diagonalized in the same basis. However, the authors argue that this criterion does not fully capture the essence of classicality. To address this gap, they propose a new framework, which can be summarized as follows:

- Given a set of m quantum states $\mathcal{E} = \rho_{x_{x=1}}^m$, with ρ_x being density operators on the Hilbert space \mathbb{C}^d , the goal is to define classicality of \mathcal{E} .
- The authors introduce classical preparation devices P_λ , labeled by a hidden variable λ , where each device prepares states diagonal in a particular basis of \mathbb{C}^d .
- A fixed probability distribution $q(\lambda)$ over these devices is assumed. For each input x and hidden variable λ , the device prepares a state $\tau_{x\lambda}$ such that $[\tau_{x\lambda}, \tau_{x'\lambda}] = 0$ for all x, x', λ .
- If, on average over λ , the correct state ρ_x is prepared for all x , then \mathcal{E} admits a classical model.

As an illustration, the authors consider the BB84 states. While these states are non-commuting, they show that under sufficient depolarizing noise a classical model exists whenever $v \leq 1/\sqrt{2}$.

The definition implies that \mathcal{E} can be simulated by a d -dimensional classical system supplemented with randomness λ . The authors further generalize this to the notion of an r -simulable set, allowing simulation by an $r < d$ -dimensional classical system. The main technical findings of the manuscript are:

- **Result 1:** For m pure states $|\psi_x\rangle$, their depolarized versions $\rho_x = v |\psi_x\rangle \langle \psi_x| + (1 - v)I_d/d$ are r -simulable whenever $v \leq \frac{H_r - 1}{d - 1}$, where $H_r = \sum_{k=1}^r 1/k$ is the harmonic number. While not optimal, this bound is an interesting analytic result.
- **Result 2:** For the set of all pure states in \mathbb{C}^d , the bound in Result 1 is both necessary and sufficient for classicality of their depolarized versions, thus making it optimal in this case.
- **Result 3:** For M mutually distinct bases, i.e. $|\psi_x\rangle_{x=1}^{dM}$, the authors obtain another analytic bound $v \leq \frac{r-1}{M(d-1)}$. Although not optimal, this may improve upon Result 1 when M is small.
- They also propose a linear programming approach to compute tighter bounds on v for depolarized sets.
- **Result 4:** The authors introduce witnesses of nonclassicality in the prepare-and-measure (PAM) scenario. Violation of these inequalities rules out classical r -simulable models. Although they mainly treat d -simulable models, their method generalizes to arbitrary r .

- **Result 5:** They highlight a connection between two-qubit steering inequalities and witnesses for qubit states ρ_x .
- **Result 6:** If ρ_x has a d -dimensional classical model, then the binary-outcome POVMs $M = \{\rho_x, I_d - \rho_x\}$ are jointly measurable. For qubits, the converse also holds.

Comments:

The manuscript presents technically correct results and introduces a new definition of classicality. Nonetheless, we find physical justification for adopting this particular definition missing. While the work has merit and could warrant publication, significant conceptual clarifications are needed. Our specific comments are:

- The authors claim that the set S of classical models is convex: if $\rho_x \in S$ and $\sigma_x \in S$, then $p\rho_x + (1-p)\sigma_x \in S$ for any $p \in [0, 1]$. This indeed follows. However, does the convex hull of a classical set necessarily remain classical? If yes, a proof should be provided; if not, a counterexample is needed.
- The manuscript does not convincingly argue why the proposed construction should be regarded as a definition of classicality. A natural operational interpretation arises in the PAM scenario: λ acts as a shared random variable; Alice, given x and λ , prepares a diagonal state labeled by z , sends z via a d -level classical system, and Bob reconstructs the state. On average, this yields ρ_x . This reasoning would justify the definition, yet the authors seem not to adopt it.

For instance, the set $|0\rangle, |+\rangle$ can be classically prepared by Alice sending one classical bit, yet this set is not classical by their definition (or since the states do not commute). Moreover, it never generates nonclassical PAM statistics. Thus, their definition diverges from the operational notion of classicality. We urge the authors to clarify what operational principle their framework captures.

- Furthermore, their construction of witnesses depend explicitly on Bob's measurement choice, unlike conventional semi-device-independent PAM witnesses such as RAC success probabilities. This suggests that the proposed framework constitutes a new paradigm where measurement settings are fixed by the witness itself. If so, the authors should clearly articulate the operational meaning of nonclassicality in this setting.
- Several section references in the Supplementary Information appear incorrect and should be carefully checked.

Suggestion:

It would be valuable to map their definition onto an operational task, analogous to how Bell inequalities certify entanglement or steering inequalities certify one-sided entanglement. Establishing such a task would give strong justification for calling these models "classical".

There is a long tradition of defining classicality through Wigner-type distribution functions [Annals of Physics 176, 1-21 (1987); J. Math. Phys. 47, 122107 (2006); J. Phys. A: Math. Gen. 39 1405 (2006) *etc*], a notion that also enjoys operational justification from computational and communication perspectives [Phys. Rev. Lett. 109, 230503 (2012); Nature 510, 351–355 (2014); arXiv.2506.19369 *etc*]. Unfortunately, the present manuscript does not discuss how the newly proposed definition offers a stronger or more compelling justification compared to this well-established approach.

In conclusion, while the results are technically correct, the manuscript in its current form does not provide a sufficiently strong conceptual foundation to warrant publication in *Nature Communications*.

Report

Title: *Operationally classical simulation of quantum states*

Authors: G. Cobucci, A. Bernal, M. J. Renner, and A. Tavakoli

In the revised manuscript the authors address our concerns with substantive clarifications and added content, particularly in connecting the proposed definition of classicality to operational Prepare-and-Measure (PAM) scenarios. While the work is technically sound and represents a meaningful contribution to quantum foundations, its scope and conceptual framing align more closely with more specialized journal such as *npj Quantum Information* rather than the broad interdisciplinary readership expected by *Nature Communications*.

Key Strengths of the Revised Manuscript

- ✓ The authors have added a new subsection and a figure explicitly linking their definition of classicality to the PAM scenario, addressing our primary concern.
- ✓ The analogy between entanglement in Bell scenarios and classicality in PAM scenarios is now well articulated, strengthening the conceptual foundation.
- ✓ The manuscript now explicitly distinguishes their notion of classicality from other frameworks (e.g., Wigner-based nonclassicality).
- ✓ The authors provide concrete examples (e.g., BB84 states and mutually unbiased bases) to show how their methods can be applied to realistic, finite sets of states—not just the full state space.
- ✓ They demonstrate how their tools can be used to witness nonclassicality in practical scenarios, which is compelling.

Reasons for Recommending npj Quant. Inf. Over Nat. Commun.

- ✓ The manuscript's core contribution is a technical framework for defining and detecting classicality in quantum preparation devices. This is a foundational and methodological advance that fits naturally within the scope of *npj Quantum Information*, which publishes focused research in quantum information science.
- ✓ While the work is important to the quantum information community, it does not present a breakthrough with immediate interdisciplinary implications or widespread technological impact—a key criterion for Nature Communications.
- ✓ The discussion of classical simulations, PAM scenarios, and dimension-dependent thresholds is highly specialized. The revised manuscript does not sufficiently reframe the work to engage a broad scientific audience beyond quantum information experts.
- ✓ Despite improvements, our original conclusion that the manuscript lacks a “sufficiently strong conceptual foundation” for Nature Communications remains valid from an interdisciplinary perspective. The work is better suited for a journal where technical depth is prioritized over broad appeal.

Suggested Minor Revisions for npj Quant. Inf. (if it is considered there)

- ✓ Authors may consider adding a short “Outlook” paragraph in the Discussion suggesting future experimental implementations of the proposed witnesses.